# [3+3]-Annulation of Cyclic Nitronates with Vinyl Diazoacetates: Diastereoselective Synthesis of Partially Saturated [1,2]Oxazino[2,3-*b*][1,2]oxazines and Their Base-Promoted Ring Contraction to Pyrrolo[1,2-*b*][1,2]oxazine Derivatives

**DOI:** 10.3390/molecules28073025

**Published:** 2023-03-28

**Authors:** Yulia A. Antonova, Yulia V. Nelyubina, Sema L. Ioffe, Andrey A. Tabolin

**Affiliations:** 1N. D. Zelinsky Institute of Organic Chemistry, Russian Academy of Sciences, Leninsky Prosp. 47, Moscow 119991, Russia; 2A. N. Nesmeyanov Institute of Organoelement Compounds, Russian Academy of Sciences, Vavilov Str. 28, Moscow 119991, Russia

**Keywords:** [3+3]-annulation, diazo compounds, nitronates, 1,2-oxazines, pyrroles, rhodium carbenoids, nitrogen heterocycles

## Abstract

A rhodium(II)-catalyzed reaction of cyclic nitronates (5,6-dihydro-4*H*-1,2-oxazine *N*-oxides) with vinyl diazoacetates proceeds as a [3+3]-annulation producing bicyclic unsaturated nitroso acetals (4a,5,6,7-tetrahydro-2*H*-[1,2]oxazino[2,3-*b*][1,2]oxazines). Optimization of reaction conditions revealed the use of Rh(II) octanoate as the preferred catalyst in THF at room temperature, which allows the preparation of target products in good yields and excellent diastereoselectivity. Under basic conditions, namely, the combined action of DBU and alcohol, these nitroso acetals undergo ring contraction of an unsaturated oxazine ring into the corresponding pyrrole. Both transformations can be performed in a one-pot fashion, thus constituting a quick approach to oxazine-annulated pyrroles from available starting materials, such as nitroalkenes, olefins, and diazo compounds.

## 1. Introduction

1,2-Oxazine derivatives represent an important class of nitrogen–oxygen heterocycles which are widely used in organic chemistry [1,2,3,4,5,6,7,8]. Reductive cleavage of the N–O bond makes 1,2-oxazine derivatives useful precursors to aminoalcohols and functionalized carbonyl compounds. Moreover, 1,2-oxazines are versatile substrates for the preparation of other important heterocyclic motifs, such as pyrrolidine, pyrrole, or furan. The recent growth of the research interest in the chemistry of 1,2-oxazines is promoted by the discovery of bioactive natural products possessing 1,2-oxazine core (Figure 1, (1)) [9,10,11,12,13,14,15,16,17,18]. Such compounds as trichodermamides or phyllantidine became attractive targets for both total synthesis and preparation of analogues in pursuit of increased bioactivity [19,20,21,22,23,24,25,26].

One of the synthetic routes to 1,2-oxazines is based on various transformations of six-membered cyclic nitronates, 5,6-dihydro-4*H*-1,2-oxazine *N*-oxides **1** (Figure 1, (2)). These substrates are readily accessible via the [4+2]-cycloaddition between such simple starting materials as conjugated nitroalkenes and olefins [27,28]. It allows a quick and stereoselective construction of an oxazine ring, while further transformations of the *N*-oxide moiety are used for the desired modifications of the assembled heterocyclic core. One of the most known and most used reactivity patterns of nitronates is the [3+2]-cycloaddition [28,29,30,31]. It leads to the assembly of annulated 6,5-bicyclic nitroso acetals, possessing six- and five-membered rings. For instance, this strategy was extensively exploited by Denmark and coworkers in the total synthesis of pyrrolizidine and indolizidine alkaloids [32,33,34,35,36,37,38,39]. In contrast to [3+2]-cycloaddition, other cycloaddition/annulation reactions of nitronates remain underexplored. Recently, we reported a formal [3+3]-cycloaddition reaction of cyclic nitronates **1** with donor–acceptor cyclopropanes (DACs) [40,41]. DACs reacted as three-carbon components and allowed a smooth diastereoselective preparation of nitroso acetals, possessing two saturated oxazine rings. To expand the scope of polycyclic frameworks, which can be available from nitronates **1**, we pursued other three-carbon annulation partners. In continuation of these studies, we turned our attention to the vinyl diazoacetates **2**, which recommended themselves as useful 1,3-dipole equivalents in annulations leading to various carbocycles (cyclobutenes, cyclopentenes, etc.) and heterocycles (pyrazoles, pyrroles, isoxazolines, quinolines, etc.) [42,43]. Herein, we report the [3+3]-annulation of nitronates **1** with vinyl diazoacetates **2** for the synthesis of 6,6-bicyclic nitroso acetals **3** and their further transformation into pyrrole-annulated oxazine derivatives.

## 2. Results and Discussion

We initiated our study with optimization of the reaction between model nitronate **1a** and vinyl diazoacetate **2a** (Table 1). Variations of catalyst, solvent, temperature, and reagent ratio were performed. Notably, only one diastereomer of adduct **3a** was detected in all experiments. Among the solvents tested, the best results were achieved in THF. In other ethereal solvents (diethyl ether, glyme, 1,4-dioxane), a decrease in yield was observed (Entries 2–4), while in acetonitrile, dichloromethane, and hexane, starting nitronate **1a** was not fully consumed (Entries 6–8). As compared to rhodium(II) octanoate, rhodium(II) catalysts (acetate, trifluoroacetate, tetramethyl-1,3-benzenedipropionate) gave worse results (Entries 10–12), while other metal catalysts that are often used in carbenoid reactions [42,44,45,46,47,48,49,50,51,52] did not produce the target product at all (Entries 13–17). Variations of reagent ratios (Entries 18–20), concentration (Entry 21), and reaction temperatures (Entries 22–23) did not lead to improvement in yield. Therefore, the use of 2 equiv. of vinyl diazoacetate and 2 mol.% of rhodium(II) octanoate in THF at r.t. (Entry 1) was chosen for subsequent studies.

With the optimized conditions in hand, an evaluation of the substrate scope was performed (Figure 2). Diazoacetates possessing various ester groups were successfully involved in the reaction. Alkyl (Me, **3b**; Et, **3c**), benzyl (**3d**), *p*-nitrobenzyl (**3e**), and phenyl (**3f**) esters gave corresponding products in good yields. However, better results were achieved for electron-accepting 2,2,2-trichloroethyl (**3a**) and *p*-nitrobenzyl (**3e**) moieties, which is in line with the previously reported advantages of these substituents in C–H functionalization and cyclopropanation [53,54,55,56]. Poor yields and/or conversions were observed for vinyl diazoacetates with bulky *tert*-butyl (**2g**) or BHT (2,6-di-*tert*-butyl-4-methylphenyl, **2h**) ester groups [57]. Unfortunately, complex mixtures were observed when employing substituted (β-phenyl or γ-phenyl) vinyl diazoacetates. Regarding the scope of nitronates **1**, the reaction tolerates various aryl groups at the C4 of starting nitronate. *Meta*-(**3i**)- and *para*-(**3a,g,h**)-substituted substrates gave products **3** smoothly, while for *ortho*-substituent (**3j**) a diminished yield was attained, presumably, due to steric hindrance. Importantly, no influence of the electronic effect of aromatic substituent was observed as electron-accepting (NO_2_, **3h**), halogen (**3g**,**j**), as well as pharmaceutically attractive dihydroxylated aromatics [58,59] (**3k**) gave comparable outcomes. Apart from aryl-, 5-benzoyloxy-(**3l**), 5-alkyl-(**3m**), and 7-ethoxy-(**3o**) products were obtained, thus demonstrating the possibility of incorporation of different functionalities in the target structures. Poor yield was attained for cyclohexane-annulated product **3p**, while a complex product mixture was observed for nitronate **1q**. When more sterically encumbered camphene-derived (**1r**) or 3-methyl-substituted nitronate (**1s**) was involved, the reaction was decelerated and starting materials were observed even after 4 days.

Structures of the obtained products were supported by their NMR (1D and 2D) and HRMS data. Gratifyingly, in all examined cases, the annulation was found to be stereoselective producing only one diastereomer of products **3**. The relative configuration of stereocenters was also deduced on the basis of the coupling constant between C(4a)–H and C(5)–H. Its high value of 9–11 Hz evidenced about (pseudo)axial positions of both protons. This conclusion was supported by the X-ray analysis for products **3b** and **3p** (Figure 1). Notably, the *cis*-junction of two oxazine rings was observed, which can be favored by anomeric interaction within O–N–O moiety [60]. However, we should mention the conformational lability of products **3**, as many signals in their NMR spectra were broadened. Especially notable was product **3p** possessing three contiguous annulated six-membered rings. In its case, a lot of the ^13^C signals were of low intensity and were assigned only with the aim of ^1^H-^13^C HSQC spectra (see Appendix A). Mentioned effects can be attributed to both inversion of six-membered rings and to nitrogen inversion [41,61].

Based on the obtained and the literature [42,43,62] data, the following mechanism for the annulation was proposed (Figure 3). In the first step, electrophilic vinyl carbenoid species **A** is generated from vinyl diazo compound **2** and a rhodium catalyst. Nucleophilic attack of the oxygen atom of *N*-oxide at the unhindered =CH_2_ terminus in **A** produces zwitterion **B** that undergoes ring closure with the expulsion of the catalyst and formation of target cycloadduct (path a). Observed stereochemistry can be explained by an approach of the bulky vinyl-rhodium moiety from the face of the dipole opposite to substituent R^1^. Hence, low yields in the case of product **3p** and the messy reaction for nitronate **1q** can be attributed to the presence of a relatively bulky substituent at C(6) of the nitronate, which together with the substituent at C(4) may block both sides of the dipole, thus preventing the annulation. Interestingly, the observed [3+3]-annulation with nitronates differs from the reaction of vinyl diazoacetates with nitrones, which often proceeds via a [3+2]-pathway and subsequent carbene reactions (cf. path b) [63,64,65,66], while the respective [3+3]-pathway was rarely observed [62].

Aiming at the subsequent functionalization of product **3b**, we attempted a Michael addition of dimethyl malonate across the C=C double bond. However, none of the desired transformation was observed, albeit ring contraction occurred providing pyrrolo[1,2-*b*]oxazine derivative **4b**. This process has some precedents in the literature [67,68,69,70,71,72,73,74,75], with the closest analogue observed by Kerr et al. for non-annulated oxazines under the treatment with DBU in MeCN [67]. The proposed mechanism starts with the deprotonation at C(2), which is facilitated by the conjugation of the formed anion with C=C–CO_2_ moiety in **D** (Figure 4). Subsequent cleavage of the N–O bond, formation of an aldehyde group, and recyclization produce hydroxypyrroline **E**, upon which dehydration furnishes the target pyrrole ring (path a). Some optimization of reaction conditions was performed (Figure 5) and a combination of base (DBU) and protic additive (alcohol) was found optimal, instead of the use of a base alone (cf. yields for DBU and DBU/TCEOH). We attribute it to two main reasons. First, N–O bonds in nitroso acetals are also susceptible to basic cleavage (e.g., Figure 4, path b) [76,77], which can lead to the formation of side products. The use of alcohol promotes the elimination of the hydroxyl group from intermediate **E** via hydrogen bonding, thus facilitating pyrrole formation (path a). Secondly, the presence of alcohol diminishes the amount of carboxylic acid **5**, which originates from the hydrolysis of product **4a** by water. Particularly, relatively large amounts of product **5** were observed when a strong base, namely, sodium *tert*-butylate was used. Similarly, in the excess of methanol, transesterification was observed (Figure 6).

The literature provides limited data on the preparation of pyrrolo[1,2-*b*]oxazine derivatives [78,79], especially possessing an unsaturated pyrrole core [80,81]. Therefore, we elucidated the scope of the found transformation. Since the reagents used for the [3+3]-annulation should not interfere with the ring contraction, we decided to make a one-pot protocol for the synthesis of pyrrolooxazines **4** from oxazine *N*-oxides **1** and vinyl diazoacetates **2**. For this purpose, the reaction mixture was evaporated after the annulation step and redissolved in CH_2_Cl_2_, followed by the addition of other reagents. This worked well providing target products **4** in reasonable yields for the whole two-step sequence (Figure 7).

## 3. Materials and Methods

### 3.1. General Experiment

All reactions were performed in oven-dried (150 °C) glassware. Most of the chemicals were acquired from commercial sources (Sigma-Aldrich, St. Louis, MO, USA; Acros Organics, Geel, Belgium; Alfa Aesar, Heysham, UK; ABCR, Karlsruhe, Germany; and P&M Invest, Moscow, Russia) with purities >95% and used as received. Petroleum ether (PE) and ethyl acetate were distilled. THF, CH_3_CN, CH_2_Cl_2_, and DBU were distilled from CaH_2_. Brine refers to the saturated aqueous solution of NaCl. TLC was performed on silica coated on aluminum with UV254 indicator. Visualization was accomplished with UV and/or anisaldehyde/H_2_SO_4_/EtOH stain. Column chromatography was performed on silica (0.04–0.063 mm, 60 Å). NMR spectra were recorded at 300 K (unless otherwise mentioned) on Bruker AM300 (Bruker, Karlsruhe, Germany), Fourier 300HD (Bruker, Karlsruhe, Germany), and Avance NEO (Bruker, Karlsruhe, Germany) spectrometers at the following spectrometer frequencies: 300 MHz (^1^H NMR) and 75 MHz (^13^C NMR). Multiplicities are assigned as s (singlet), d (doublet), t (triplet), q (quartet), m (multiplet), br (broad), and app (apparent). Assignment (including rel. configuration) was made using 2D NMR spectra for selected products. For other products, the assignment was made by analogy. High-resolution mass spectra were acquired on Bruker micrOTOF (Bruker, Bremen, Germany) spectrometer using electrospray ionization (ESI). Melting points were determined on a Koffler melting point apparatus and are uncorrected. Starting nitroalkenes [82,83,84,85,86], nitronates **1** [40,87,88,89], and vinyl diazoacetates **2** [55,90,91,92,93,94,95,96,97] were prepared according to the literature (see Appendix A for detailed procedures).

### 3.2. X-ray Crystallography

X-ray diffraction data for **3b**, **3p**, and **4b** were collected at 100 K with a Bruker Quest D8 CMOS diffractometer (Bruker, Karlsruhe, Germany), using graphite-monochromated Mo-Kα radiation (λ = 0.71073 Å, ω-scans). Structures were solved using Intrinsic Phasing with the ShelXT [98] structure solution program in Olex2 [99] and then refined with the XL refinement package [100] using Least-Squares minimization against F^2^ in the anisotropic approximation for non-hydrogen atoms. Positions of other hydrogen atoms were calculated, and they were refined in the isotropic approximation within the riding model. Crystal data and structure refinement parameters are given in Appendix A (see Appendix A). The crystallographic information for compounds **3b**, **3p**, and **4b** was deposited in the Cambridge Crystallographic Data Centre (CCDC 2242307-2242309) and can be obtained free of charge via https://www.ccdc.cam.ac.uk/structures/ (accessed on 25 February 2023).

### 3.3. Synthetic Procedures for Products ***3***–***5***

#### 3.3.1. General Procedure for the Synthesis of Nitroso Acetals **3** (GP-1)

To the 0.3 M solution of nitronate **1** (1 equiv.) in THF, Rh_2_(Oct)_4_ (0.02 equiv.) was added under an argon atmosphere at r.t. and the mixture was stirred for 5 min. Then THF solution of vinyl diazoacetate **2** (2 equiv. rel. to nitronate **1**, C = 0.5 M) was added dropwise (approx. 5–10 min.) using a syringe with stirring. The resulting solution was left overnight and then evaporated. The residue was preadsorbed onto Celite^®^ or silica gel and subjected to column chromatography on silica gel (eluent: PE/EtOAc or PE/CH_2_Cl_2_) to give target nitroso acetal **3**.

*2,2,2-Trichloroethyl (4aR*,5S*)-5-(4-methoxyphenyl)-7,7-dimethyl-4a,5,6,7-tetrahydro-2H-[1,2]oxazino[2,3-b][1,2]oxazine-4-carboxylate* **3a**. Nitroso acetal **3a** was obtained from nitronate **1a** (24 mg, 0.10 mmol) and vinyl diazoacetate **2a** (50 mg, 0.21 mmol) according to GP-1. Column chromatography (eluent: PE/EtOAc, 10:1) afforded 39 mg (85%) of the target nitroso acetal **3a** as colorless oil, which solidified upon storage in a fridge (*ca*. 4 °C). *R*_f_ = 0.30 (PE/EtOAc, 5:1, UV, anisaldehyde). m.p. = 112–114 °C (EtOAc). ^1^H NMR (300 MHz, CDCl_3_): δ 1.38 (s, 3H, Me(7)), 1.63 (s, 3H, Me(7)), 1.92 (dd, *J* = 13.3, 3.1 Hz, 1H, CH_2eq_(6)), 2.24 (app t, *J* = 12.9 Hz, 1H, CH_2ax_(6)), 3.50 (ddd, *J* = 13.4, 10.5, 3.1 Hz, 1H, CH(5)), 3.80 (s, 3H, OMe), 4.05–4.09 (m, 2H, OCH_2a_CCl_3_ and CH(4a)), 4.39 (d, *J* = 11.9 Hz, 1H, OCH_2b_CCl_3_), 4.55 (dd, *J* = 18.1, 3.5 Hz, 1H, CH_2a_(2)), 4.80 (br d, *J* = 18.1 Hz, 1H, CH_2b_(2)), 6.83 (d, *J* = 8.7 Hz, 2H, CH_Ar_), 6.95 (dd, *J* = 3.2, 1.5 Hz, 1H, CH(3)), 7.18 (d, *J* = 8.7 Hz, 2H, CH_Ar_). ^13^C NMR (75 MHz, DEPT, CDCl_3_): δ 28.7 (Me(7)), 30.8 (Me(7)), 36.3 (CH(5)), 41.2 (CH_2_(6)), 55.3 (OMe), 65.1 (CH(4a)), 68.5 (CH_2_(2)), 73.8 (OCH_2_CCl_3_), 79.1 (C(7)), 94.7 (CCl_3_), 113.7 (CH_Ar_), 129.6 (CH_Ar_), 130.4 and 132.0 (C(4) and CH_Ar_), 138.6 (=CH(3)), 158.7 (C_Ar_–OMe), 162.8 (C=O). HRMS (ESI): *m*/*z* calcd. for C_19_H_23_Cl_3_NO_5_^+^ [M + H]^+^: 450.0636, found: 450.0649.

*Methyl (4aR*,5S*)-5-(4-methoxyphenyl)-7,7-dimethyl-4a,5,6,7-tetrahydro-2H-[1,2]oxazino[2,3-b][1,2]oxazine-4-carboxylate* **3b**. Nitroso acetal **3b** was obtained from nitronate **1a** (24 mg, 0.10 mmol) and vinyl diazoacetate **2b** (50 mg, 0.21 mmol) according to GP-1. Column chromatography (eluent: PE/EtOAc, 10:1) afforded 20 mg (59%) of the target nitroso acetal **3b** as colorless oil, which solidified upon storage in a fridge (*ca*. 4 °C). *R*_f_ = 0.59 (PE/EtOAc, 1:1, UV, anisaldehyde). m.p. = 131–133 °C (PE/CH_2_Cl_2_, 3:1). ^1^H NMR (300 MHz, COSY, CDCl_3_): δ 1.37 (s, 3H, Me_eq_(7)), 1.62 (s, 3H, Me_ax_(7)), 1.90 (dd, *J* = 13.3, 3.1 Hz, 1H, CH_2eq_(6)), 2.24 (app t, *J* = 13.1 Hz, 1H, CH_2ax_(6)), 3.19 (s, 3H, CO_2_Me), 3.47 (ddd, *J* = 12.8, 10.5, 2.7 Hz, 1H, CH(5)), 3.80 (s, 3H, OMe), 4.01 (dd, *J* = 10.5, 1.5 Hz, 1H, CH(4a)), 4.49 (dd, *J* = 17.8, 3.3 Hz, 1H, CH_2a_(2)), 4.75 (br d, *J* = 17.8 Hz, 1H, CH_2b_(2)), 6.77 (dd, *J* = 3.3, 1.4 Hz, 1H, CH(3)), 6.84 (d, *J* = 8.7 Hz, 2H, CH_Ar_), 7.16 (d, *J* = 8.7 Hz, 2H, CH_Ar_). Characteristic NOESY interactions: CH(5)/CH_2eq_(6); CH_2ax_(6)/CH(4a); CH_2ax_(6)/CH_Ar_; CH(4a)/CH_Ar_; Me_ax_(7)/CH(5). ^13^C NMR (75 MHz, DEPT, HSQC, HMBC, CDCl_3_): δ 28.6 (Me(7)), 30.9 (Me(7)), 36.3 (CH(5)), 41.1 (CH_2_(6)), 51.5 (CO_2_Me), 55.3 (OMe), 65.4 (CH(4a)), 68.5 (CH_2_(2)), 78.9 (C(7)), 113.6 (CH_Ar_), 129.4 (CH_Ar_), 131.6 and 132.3 (C(4) and CH_Ar_), 136.4 (=CH(3)), 158.6 (C_Ar_–OMe), 165.4 (C=O). HRMS (ESI): *m*/*z* calcd. for C_18_H_24_NO_5_^+^ [M + H]^+^: 334.1649, found: 334.1645. The crystallographic information for compound **3b** was deposited in the Cambridge Crystallographic Data Centre (CCDC 2242307).

*4-Ethyl (4aR*,5S*)-5-(4-methoxyphenyl)-7,7-dimethyl-4a,5,6,7-tetrahydro-2H-[1,2]oxazino[2,3-b][1,2]oxazine-4-carboxylate* **3c**. Nitroso acetal **3c** was obtained from nitronate **1a** (21 mg, 0.09 mmol) and vinyl diazoacetate **2c** (25 mg, 0.18 mmol) according to GP-1. Column chromatography (eluent: PE/EtOAc, 10:1, then 8:1) afforded 18 mg (58%) of the target nitroso acetal **3c** as colorless oil, which solidified upon storage in a fridge (*ca*. 4 °C). *R*_f_ = 0.30 (PE/EtOAc, 3:1, UV, anisaldehyde). m.p. = 132–134 °C (PE/CH_2_Cl_2_, 3:1). ^1^H NMR (300 MHz, CDCl_3_): δ 1.02 (t, *J* = 7.1 Hz, 3H, OCH_2_CH_3_), 1.37 (s, 3H, Me(7)), 1.62 (s, 3H, Me(7)), 1.90 (dd, *J* = 13.2, 3.0 Hz, 1H, CH_2eq_(6)), 2.23 (app t, *J* = 13.1 Hz, 1H, CH_2ax_(6)), 3.39–3.35 (m, 2H, CH(5) and OCH_2a_CH_3_), 3.72–3.83 (m, 1H, OCH_2b_CH_3_), 3.79 (s, overlapped, 3H, OMe), 4.03 (dd, *J* = 10.6, 1.4 Hz, 1H, CH(4a)), 4.50 (dd, *J* = 17.7, 3.4 Hz, 1H, CH_2a_(2)), 4.75 (br d, *J* = 17.7 Hz, 1H, CH_2b_(2)), 6.75 (d, *J* = 3.1, 1.4 Hz, 1H, CH(3)), 6.83 (d, *J* = 8.7 Hz, 2H, CH_Ar_), 7.16 (d, *J* = 8.7 Hz, 2H, CH_Ar_). ^13^C NMR (75 MHz, DEPT, CDCl_3_): δ 13.9 (OCH_2_CH_3_), 28.6 (Me(7)), 30.9 (Me(7)), 36.3 (CH(5)), 41.2 (CH_2_(6)), 55.3 (OMe), 60.6 (OCH_2_CH_3_), 65.3 (CH(4a)), 68.4 (CH_2_(2)), 79.0 (C(7)), 113.6 (CH_Ar_), 129.5 (CH_Ar_), 131.9 and 132.3 (C_Ar_ and C(4)), 135.9 (=CH(3)), 158.7 (C_Ar_–O), 165.0 (C=O). HRMS (ESI): *m*/*z* calcd. for C_19_H_26_NO_5_^+^ [M + H]^+^: 348.1805, found: 348.1805.

*Benzyl (4aR*,5S*)-5-(4-methoxyphenyl)-7,7-dimethyl-4a,5,6,7-tetrahydro-2H-[1,2]oxazino[2,3-b][1,2]oxazine-4-carboxylate* **3d**. Nitroso acetal **3d** was obtained from nitronate **1a** (29 mg, 0.13 mmol) and vinyl diazoacetate **2d** (47 mg, 0.25 mmol) according to GP-1. Column chromatography (eluent: PE/EtOAc, 15:1) afforded 37 mg (73%) of the target nitroso acetal **3d** as slightly yellow oil. Analytically pure material was obtained after crystallization from PE/CH_2_Cl_2_ (1:1) as white powder. *R*_f_ = 0.62 (PE/EtOAc, 1:1, UV, anisaldehyde). m.p. = 110–112 °C (PE/CH_2_Cl_2_, 1:1). ^1^H NMR (300 MHz, CDCl_3_): δ 1.36 (s, 3H, Me(7)), 1.62 (s, 3H, Me(7)), 1.90 (dd, *J* = 13.3, 3.1 Hz, 1H, CH_2eq_(6)), 2.22 (app t, *J* = 13.1 Hz, 1H, CH_2ax_(6)), 3.49 (ddd, *J* = 13.3, 10.5, 3.1 Hz, 1H, CH(5)), 3.80 (s, 3H, OMe), 4.07 (app d, *J* = 10.3 Hz, 1H, CH(4a)), 4.38 (d, *J* = 12.5 Hz, 1H, OCH_2a_Ph), 4.50 (dd, *J* = 17.8, 3.5 Hz, 1H, CH_2a_(2)), 4.73–4.82 (m, 2H, CH_2b_(2) and OCH_2b_Ph), 6.78–6.81 (m, 3H, CH_Ar_ and CH(3)), 7.14–7.19 (m, 4H, CH_Ar_), 7.32–7.38 (m, 3H, CH_Ar_). ^13^C NMR (75 MHz, DEPT, CDCl_3_): δ 28.7 (Me(7)), 30.9 (Me(7)), 36.3 (CH(5)), 41.2 (CH_2_(6)), 55.3 (OMe), 65.3 (CH(4a)), 66.2 (OCH_2_Ph), 68.5 (CH_2_(2)), 79.0 (C(7)), 113.7 (CH_Ar_), 128.0 (CH_Ph_), 128.1 (CH_Ph_), 128.4 (CH_Ph_), 129.5 (CH_Ar_), 131.6 and 132.2 (C_Ar_ and C(4)), 135.6 (C_Ph_), 136.6 (=CH(3)), 158.6 (C_Ar_–O), 164.7 (C=O). HRMS (ESI): *m*/*z* calcd. for C_24_H_28_NO_5_^+^ [M + H]^+^: 410.1962, found: 410.1964.

*4-Nitrobenzyl (4aR*,5S*)-5-(4-methoxyphenyl)-7,7-dimethyl-4a,5,6,7-tetrahydro-2H-[1,2]oxazino[2,3-b][1,2]oxazine-4-carboxylate* **3e**. Nitroso acetal **3e** was obtained from nitronate **1a** (35 mg, 0.15 mmol) and vinyl diazoacetate **2e** (73 mg, 0.30 mmol) according to GP-1. Column chromatography (eluent: PE/EtOAc, 7:1, then 5:1) afforded 48 mg (72%) of the target nitroso acetal **3e** as colorless oil, which solidified upon storage in a fridge (*ca*. 4 °C). *R*_f_ = 0.52 (PE/EtOAc, 1:1, UV, anisaldehyde). m.p. = 140–142 °C (EtOAc). ^1^H NMR (300 MHz, CDCl_3_): δ 1.37 (s, 3H, Me(7)), 1.62 (s, 3H, Me(7)), 1.90 (dd, *J* = 13.3, 2.9 Hz, 1H, CH_2eq_(6)), 2.21 (app t, *J* = 13.1 Hz, 1H, CH_2ax_(6)), 3.49 (ddd, *J* = 13.3, 10.5, 3.1 Hz, 1H, CH(5)), 3.76 (s, 3H, OMe), 4.06 (app d, *J* = 10.6 Hz, 1H, CH(4a)), 4.46 (d, *J* = 13.5 Hz, 1H, OCH_2a_Ar), 4.53 (dd, *J* = 18.0, 3.5 Hz, 1H, CH_2a_(2)), 4.78 (d, *J* = 18.0 Hz, 1H, CH_2b_(2)), 4.93 (d, *J* = 13.5 Hz, 1H, OCH_2b_Ar), 6.73 (d, *J* = 8.6 Hz, 2H, CH_ArOMe_), 6.87 (br s, 1H, CH(3)), 7.13 (d, *J* = 8.6 Hz, 2H, CH_ArOMe_), 7.29 (d, *J* = 8.7 Hz, 2H, CH_ArNO2_), 8.20 (d, *J* = 8.7 Hz, 2H, CH_ArNO2_). ^13^C NMR (75 MHz, DEPT, HSQC, CDCl_3_): δ 28.6 (Me(7)), 30.8 (Me(7)), 36.3 (CH(5)), 41.2 (CH_2_(6)), 55.2 (OMe), 64.8 (OCH_2_Ar), 65.2 (CH(4a)), 68.5 (CH_2_(2)), 79.1 (C(7)), 113.6 (CH_ArOMe_), 123.7 (CH_ArNO2_), 128.2 (CH_ArNO2_), 129.5 (CH_ArOMe_), 131.1 and 132.2 (C_ArOMe_ and C(4)), 137.7 (=CH(3)), 142.9 (C_ArNO2_), 147.6 (C_Ar_–NO_2_), 158.6 (C_Ar_–O), 164.4 (C=O). HRMS (ESI): *m*/*z* calcd. for C_24_H_27_N_2_O_7_^+^ [M + H]^+^: 455.1813, found: 455.1807.

*Phenyl (4aR*,5S*)-5-(4-methoxyphenyl)-7,7-dimethyl-4a,5,6,7-tetrahydro-2H-[1,2]oxazino[2,3-b][1,2]oxazine-4-carboxylate* **3f**. Nitroso acetal **3f** was obtained from nitronate **1a** (30 mg, 0.13 mmol) and vinyl diazoacetate **2f** (47 mg, 0.25 mmol) according to GP-1. Column chromatography (eluent: PE/EtOAc, 20:1, then 9:1) afforded 68 mg of crude nitroso acetal, which was crystallized from PE/CH_2_Cl_2_, 1:1 to give 34 mg (68%) of the target nitroso acetal **3f** as white powder. *R*_f_ = 0.30 (PE/EtOAc, 3:1, UV, anisaldehyde). m.p. = 175–176 °C (PE/CH_2_Cl_2_, 1:1). ^1^H NMR (300 MHz, CDCl_3_): δ 1.39 (s, 3H, Me(7)), 1.66 (s, 3H, Me(7)), 1.95 (dd, *J* = 13.2, 3.1 Hz, 1H, CH_2eq_(6)), 2.26 (app t, *J* = 13.1 Hz, 1H, CH_2ax_(6)), 3.58 (ddd, *J* = 13.3, 10.5, 3.0 Hz, 1H, CH(5)), 3.79 (s, 3H, OMe), 4.18 (dd, *J* = 10.5, 1.9 Hz, 1H, CH(4a)), 4.59 (dd, *J* = 18.0, 3.4 Hz, 1H, CH_2a_(2)), 4.85 (dt, *J* = 18.0, 1.9 Hz, 1H, CH_2b_(2)), 6.54–6.59 (m, 2H, CH_Ph_), 6.85 (d, *J* = 8.7 Hz, 2H, CH_Ar_), 7.01 (dd, *J* = 3.4, 1.9 Hz, 1H, CH(3)), 7.12–7.18 (m, 1H, CH_Ph_), 7.22–7.28 (m, 4H, CH_Ar_ and CH_Ph_). ^13^C NMR (75 MHz, DEPT, CDCl_3_): δ 28.7 (Me(7)), 30.9 (Me(7)), 36.4 (CH(5)), 41.4 (CH_2_(6)), 55.3 (OMe), 65.2 (CH(4a)), 68.6 (CH_2_(2)), 79.2 (C(7)), 114.1 (CH_Ar_), 121.3 (CH_Ph_), 126.6 (CH_Ph_), 129.0 and 129.6 (CH_Ar_ and CH_Ph_), 131.2 and 132.2 (C_Ar_ and C(4)), 138.1 (=CH(3)), 150.2 (C_Ph_–O), 158.8 (C_Ar_–O), 163.1 (C=O). HRMS (ESI): *m*/*z* calcd. for C_23_H_26_NO_5_^+^ [M + H]^+^: 396.1805, found: 396.1806.

*2,2,2-Trichloroethyl (4aR*,5S*)-5-(4-chlorophenyl)-7,7-dimethyl-4a,5,6,7-tetrahydro-2H-[1,2]oxazino[2,3-b][1,2]oxazine-4-carboxylate* **3g**. Nitroso acetal **3g** was obtained from nitronate **1b** (35 mg, 0.15 mmol) and vinyl diazoacetate **2a** (72 mg, 0.30 mmol) according to GP-1. Column chromatography (eluent: PE/CH_2_Cl_2_, 1:1, then CH_2_Cl_2_) afforded 54 mg (80%) of the target nitroso acetal **3g** as colorless oil, which solidified upon storage in a fridge (*ca*. 4 °C). *R*_f_ = 0.48 (PE/EtOAc, 3:1, UV, anisaldehyde). m.p. = 121–123 °C (PE/CH_2_Cl_2_, 3:1). ^1^H NMR (300 MHz, CDCl_3_): δ 1.38 (s, 3H, Me(7)), 1.63 (s, 3H, Me(7)), 1.93 (dd, *J* = 13.2, 3.0 Hz, 1H, CH_2eq_(6)), 2.24 (app t, *J* = 13.2 Hz, 1H, CH_2ax_(6)), 3.55 (ddd, *J* = 13.3, 10.5, 3.1 Hz, 1H, CH(5)), 4.06–4.12 (m, overlapped, 1H, CH(4a)), 4.13 (d, overlapped, *J* = 11.9 Hz, 1H, OCH_2a_CCl_3_), 4.42 (d, *J* = 11.9 Hz, 1H, OCH_2b_CCl_3_), 4.56 (dd, *J* = 18.2, 3.5 Hz, 1H, CH_2a_(2)), 4.81 (app d, *J* = 18.2 Hz, 1H, CH_2b_(2)), 7.00 (dd, *J* = 3.5, 1.7 Hz, 1H, CH(3)), 7.20–7.28 (m, 4H, CH_Ar_). ^13^C NMR (75 MHz, DEPT, CDCl_3_): δ 28.6 (Me(7)), 30.8 (Me(7)), 36.8 (CH(5)), 40.9 (CH_2_(6)), 64.7 (CH(4a)), 68.5 (CH_2_(2)), 73.8 (OCH_2_CCl_3_), 78.9 (C(7)), 94.6 (CCl_3_), 128.5 (CH_Ar_), 130.0 (CH_Ar_ and C(4)), 133.0 (C_Ar_), 138.5 (C_Ar_), 139.3 (=CH(3)), 162.7 (C=O). HRMS (ESI): *m*/*z* calcd. for [C_18_H_20_Cl_4_NO_4_^+^ [M + H]^+^: 454.0141, found: 456.0145.

*2,2,2-Trichloroethyl (4aR*,5S*)-7,7-dimethyl-5-(4-nitrophenyl)-4a,5,6,7-tetrahydro-2H-[1,2]oxazino[2,3-b][1,2]oxazine-4-carboxylate* **3h**. Nitroso acetal **3h** was obtained from nitronate **1c** (27 mg, 0.11 mmol) and vinyl diazoacetate **2a** (55 mg, 0.23 mmol) according to GP-1. Column chromatography (eluent: PE/EtOAc, 20:1, then 15:1) afforded 38 mg (78%) of the target nitroso acetal **3h** as colorless oil. *R*_f_ = 0.36 (PE/EtOAc, 3:1, UV, anisaldehyde). NB: Even on short standing in solution in CDCl_3_, decomposition of compound **3h** was observed. ^1^H NMR (300 MHz, CDCl_3_): δ 1.40 (s, 3H, Me(7)), 1.64 (s, 3H, Me(7)), 1.96 (dd, *J* = 13.2, 2.6 Hz, 1H, CH_2eq_(6)), 2.30 (app t, *J* = 13.1 Hz, 1H, CH_2ax_(6)), 3.71 (br t, *J* = 12.3 Hz, 1H, CH(5)), 4.14 (br d, *J* = 10.7 Hz, 1H, CH(4a)), 4.22 (d, *J* = 11.9 Hz, 1H, OCH_2a_CCl_3_), 4.33 (d, *J* = 11.9 Hz, 1H, OCH_2b_CCl_3_), 4.59 (dd, *J* = 18.3, 3.2 Hz, 1H, CH_2a_(2)), 4.83 (app d, *J* = 18.3 Hz, 1H, CH_2b_(2)), 7.09 (br s, 1H, CH(3)), 7.46 (d, *J* = 8.6 Hz, 2H, CH_Ar_), 8.15 (d, *J* = 8.7 Hz, 2H, CH_Ar_). ^13^C NMR (75 MHz, DEPT, CDCl_3_): δ 28.6 (Me(7)), 30.8 (Me(7)), 37.5 (CH(5)), 40.8 (CH_2_(6)), 64.1 (CH(4a)), 68.6 (CH_2_(2)), 73.8 (OCH_2_CCl_3_), 78.7 (C(7)), 94.5 (CCl_3_), 123.4 (CH_Ar_), 129.1 (C(4)), 129.6 (CH_Ar_), 140.4 (=CH(3) and C_Ar_), 147.1 (C_Ar_), 162.5 (C=O). HRMS (ESI): *m*/*z* calcd. for C_18_H_20_Cl_4_NO_4_^+^ [M + H]^+^: 454.0141, found: 456.0145.

*2,2,2-Trichloroethyl (4aR*,5S*)-5-(3-methoxyphenyl)-7,7-dimethyl-4a,5,6,7-tetrahydro-2H-[1,2]oxazino[2,3-b][1,2]oxazine-4-carboxylate* **3i**. Nitroso acetal **3i** was obtained from nitronate **1d** (33 mg, 0.14 mmol) and vinyl diazoacetate **2a** (68 mg, 0.28 mmol) according to GP-1. Column chromatography (eluent: PE/EtOAc, 20:1, then 15:1) afforded 47 mg (76%) of the target nitroso acetal **3i** as slightly yellow oil, which solidified upon storage in a fridge (*ca*. 4 °C). *R*_f_ = 0.30 (PE/EtOAc, 5:1, UV, anisaldehyde). m.p. = 107–109 °C (PE/CH_2_Cl_2_, 2:1). ^1^H NMR (300 MHz, CDCl_3_): δ 1.38 (s, 3H, Me(7)), 1.63 (s, 3H, Me(7)), 1.95 (dd, *J* = 13.3, 3.1 Hz, 1H, CH_2eq_(6)), 2.28 (app t, *J* = 13.1 Hz, 1H, CH_2ax_(6)), 3.54 (ddd, *J* = 13.3, 10.5, 3.1 Hz, 1H, CH(5)), 3.81 (s, 3H, OMe), 4.03 (d, *J* = 11.9 Hz, 1H, OCH_2a_CCl_3_), 4.10 (d, *J* = 10.5 Hz, 1H, CH(4a)), 4.40 (d, *J* = 11.9 Hz, 1H, OCH_2b_CCl_3_), 4.56 (dd, *J* = 18.1, 3.5 Hz, 1H, CH_2a_(2)), 4.81 (app d, *J* = 18.1 Hz, 1H, CH_2b_(2)), 6.76–6.81 (m, 2H, CH_Ar_), 6.87 (d, *J* = 7.7 Hz, 1H, CH_Ar_), 6.97 (dd, *J* = 3.5, 1.7 Hz, 1H, CH(3)), 7.22 (app t, *J* = 7.7 Hz, 1H, CH_Ar_). ^13^C NMR (75 MHz, DEPT, CDCl_3_): δ 28.6 (Me(7)), 30.8 (Me(7)), 37.2 (CH(5)), 40.7 (CH_2_(6)), 55.2 (OMe), 64.9 (CH(4a)), 68.5 (CH_2_(2)), 73.7 (OCH_2_CCl_3_), 79.0 (C(7)), 94.7 (CCl_3_), 112.5 (CH_Ar_), 114.5 (CH_Ar_), 120.9 (CH_Ar_), 129.4 (CH_Ar_), 130.4 (C(4)), 138.6 (=CH(3)), 141.6 (C_Ar_), 159.6 (C_Ar_–OMe), 162.8 (C=O). HRMS (ESI): *m*/*z* calcd. for C_19_H_23_Cl_3_NO_5_^+^ [M + H]^+^: 450.0636, found: 450.0642.

*2,2,2-Trichloroethyl (4aR*,5S*)-5-(2-chlorophenyl)-7,7-dimethyl-4a,5,6,7-tetrahydro-2H-[1,2]oxazino[2,3-b][1,2]oxazine-4-carboxylate* **3j**. Nitroso acetal **3j** was obtained from nitronate **1e** (28 mg, 0.12 mmol) and vinyl diazoacetate **2a** (57 mg, 0.24 mmol) according to GP-1. Column chromatography (eluent: PE/EtOAc, 40:1) afforded 32 mg (60%) of the target nitroso acetal **3j** as colorless oil, which solidified upon storage in a fridge (*ca*. 4 °C). *R*_f_ = 0.51 (PE/EtOAc, 3:1, UV, anisaldehyde). m.p. = 147–149 °C (PE/CH_2_Cl_2_, 2:1). ^1^H NMR (300 MHz, CDCl_3_): δ 1.38 (s, 3H, Me(7)), 1.67 (br s, 3H, Me(7)), 1.88 (dd, *J* = 13.2, 2.7 Hz, 1H, CH_2eq_(6)), 2.22 (app t, *J* = 13.0 Hz, 1H, CH_2ax_(6)), 4.10–4.16 (m, 2H, CH(4a) and OCH_2a_CCl_3_), 4.36–4.46 (m, 2H, CH(5) and OCH_2b_CCl_3_), 4.60 (dd, *J* = 18.2, 3.3 Hz, 1H, CH_2a_(2)), 4.81 (app d, *J* = 18.6 Hz, 1H, CH_2b_(2)), 7.00 (br s, 1H, CH(3)), 7.13–7.19 (m, 1H, CH_Ar_), 7.27–7.32 (m, 2H, CH_Ar_), 7.13–7.44–7.47 (m, 1H, CH_Ar_). ^13^C NMR (75 MHz, DEPT, CDCl_3_): δ 28.7 (Me(7)), 30.7 (Me(7)), 32.1 (br, CH(5)), 41.2 (CH_2_(6)), 64.5 (CH(4a)), 68.5 (CH_2_(2)), 73.7 (OCH_2_CCl_3_), 78.9 (C(7)), 94.7 (CCl_3_), 126.9 (CH_Ar_), 128.1 (CH_Ar_), 129.1 (C(4)), 129.5 (CH_Ar_), 129.8 (CH_Ar_), 134.4 (C_Ar_), 137.6 (C_Ar_), 139.6 (=CH(3)), 162.8 (C=O). HRMS (ESI): *m*/*z* calcd. for C_18_H_20_Cl_4_NO_4_^+^ [M + H]^+^: 454.0141, found: 454.0136.

*2,2,2-Trichloroethyl (4aR*,5S*)-5-(3-(cyclopentyloxy)-4-methoxyphenyl)-7,7-dimethyl-4a,5,6,7-tetrahydro-2H-[1,2]oxazino [2,3-b][1,2]oxazine-4-carboxylate* **3k**. Nitroso acetal **3k** was obtained from nitronate **1f** (36 mg, 0.11 mmol) and vinyl diazoacetate **2a** (54 mg, 0.22 mmol) according to GP-1. Column chromatography (eluent: PE/EtOAc, 20:1, then 15:1) afforded 44 mg (73%) of the target nitroso acetal **3k** as colorless oil. *R*_f_ = 0.46 (PE/EtOAc, 3:1, UV, anisaldehyde). ^1^H NMR (300 MHz, CDCl_3_): δ 1.38 (s, 3H, Me(7)), 1.63 (br s, 3H, Me(7) and CH_2cycl_), 1.80–2.01 (m, 7H, 3 × CH_2cycl_ and CH_2eq_(5)), 2.21 (app t, *J* = 13.1 Hz, 1H, CH_2ax_(6)), 3.46 (ddd, *J* = 13.3, 10.5, 3.1 Hz, 1H, CH(5)), 3.83 (s, 3H, OMe), 4.07 (dd, *J* = 10.5, 1.9 Hz, 1H, CH(4a)), 4.16 (d, *J* = 12.0 Hz, 1H, OCH_2a_CCl_3_), 4.37 (d, *J* = 12.0 Hz, 1H, OCH_2b_CCl_3_), 4.56 (dd, *J* = 18.0, 3.3 Hz, 1H, CH_2a_(2)), 4.77–4.83 (m, 2H, CH_2b_(2) and CH_cycl_–O), 6.75–6.78 (m, 3H, CH_Ar_), 6.94 (dd, *J* = 3.3, 1.4 Hz, 1H, CH(3)). ^13^C NMR (75 MHz, DEPT, CDCl_3_): δ 24.1 and 24.2 (2 × CH_2cycl_), 28.7 (Me(7)), 30.9 (Me(7)), 32.88 and 32.92 (2 × CH_2cycl_), 36.7 (CH(5)), 41.1 (CH_2_(6)), 56.2 (OMe), 65.1 (CH(4a)), 68.5 (CH_2_(2)), 73.7 (OCH_2_CCl_3_), 79.1 (C(7)), 80.5 (CH_cycl_–O), 94.8 (CCl_3_), 111.8 (CH_Ar_), 115.5 (CH_Ar_), 120.6 (CH_Ar_), 130.6 (C(4)), 132.4 (C_Ar_), 138.2 (=CH(3)), 147.6 (C_Ar_–O), 149.2 (C_Ar_–O), 162.6 (C=O). HRMS (ESI): *m*/*z* calcd. for C_24_H_31_Cl_3_NO_6_^+^ [M + H]^+^: 534.1211, found: 534.1196.

*2,2,2-Trichloroethyl (4aS*,5R*)-5-(benzoyloxy)-7,7-dimethyl-4a,5,6,7-tetrahydro-2H-[1,2]oxazino[2,3-b][1,2]oxazine-4-carboxylate* **3l**. Nitroso acetal **3l** was obtained from nitronate **1g** (34 mg, 0.14 mmol) and vinyl diazoacetate **2a** (68 mg, 0.28 mmol) according to GP-1. Column chromatography (eluent: PE/EtOAc, 10:1) afforded 48 mg of crude nitroso acetal **3l** as colorless oil. Then, column chromatography (eluent: PE/CH_2_Cl_2_, 1:1, then CH_2_Cl_2_) afforded 40 mg (63%) of the target nitroso acetal **3l** as colorless oil. *R*_f_ = 0.50 (PE/EtOAc, 3:1, UV, anisaldehyde). ^1^H NMR (300 MHz, 320 K, COSY, CDCl_3_): δ 1.42 (s, 3H, Me), 1.63 (br s, 3H, Me), 1.98 (dd, *J* = 12.8, 10.3 Hz, 1H, CH_2a_(6)), 2.20–2.40 (br m, 1H, CH_2b_(6)), 4.29 (d, *J* = 9.3 Hz, 1H, CH(4a)), 4.52–4.85 (m, 4H, OCH_2_CCl_3_ and CH_2_(2)), 5.80–5.91 (br m, 1H, CH(5)), 7.11–7.22 (m, 1H, CH(3)), 7.46 (app t, *J* = 7.5 Hz, 2H, CH_Ph_), 7.58 (app tt, *J* = 7.4, 2.1 Hz, 1H, CH_Ph_), 8.05 (app d, *J* = 7.2 Hz, 2H, CH_Ph_). ^13^C NMR (75 MHz, 320 K, DEPT, HSQC, HMBC, CDCl_3_): δ 29.3 (Me), 30.4 (br, Me), 40.4 (br, CH_2_(6)), 62.0 (CH(4a)), 67.9 (CH(5)), 74.1 (OCH_2_CCl_3_), 81.2 (CH(7)), 94.7 (CCl_3_), 128.4 (CH_Ph_), 129.8 (CH_Ph_ and C_Ph_), 133.2 (CH_Ph_), 140.0 (=CH(3)), 162.9 (C=O), 165.8 (C=O). CH_2_(2) and C(4) were not observed due to broadening/low intensity. ^1^H NMR (300 MHz, 343 K, COSY, DMSO-*d_6_*): δ 1.35 (s, 3H, Me_eq_), 1.50 (br s, 3H, Me_ax_), 1.95 (dd, *J* = 13.0, 9.7 Hz, 1H, CH_2ax_(6)), 2.15–2.27 (br m, 1H, CH_2eq_(6)), 4.15 (d, *J* = 8.3 Hz, 1H, CH(4a)), 4.62–4.65 (m, 2H, CH_2_(2)), 4.77 (s, 2H, OCH_2_CCl_3_), 5.77 (app td, *J* = 9.1, 4.1 Hz, 1H, CH(5)), 7.20 (br s, 1H, CH(3)), 7.53 (app t, *J* = 7.6 Hz, 2H, CH_Ph_), 7.67 (app tt, *J* = 7.4, 1.3 Hz, 1H, CH_Ph_), 7.95 (app d, *J* = 7.1 Hz, 2H, CH_Ph_). Characteristic NOESY interactions: Me_ax_/CH(5); CH_2eq_(6)/CH(5); CH_2ax_(6)/CH(4a); Me_eq_/CH_2ax_(6). ^13^C NMR (75 MHz, 343 K, DEPT, HSQC, HMBC, DMSO-d6): δ 29.6 (2×Me), 39.8 (CH_2_(6)), 61.9 (CH(4a)), 67.3 (br, CH_2_(2)), 68.3 (CH(5)), 74.0 (OCH_2_CCl_3_), 80.6 (br, CH(7)), 95.5 (CCl_3_), 126.8 (br, C(4)), 129.1 (CH_Ph_), 129.7 (CH_Ph_), 130.0 (C_Ph_), 133.9 (CH_Ph_), 141.4 (=CH(3)), 163.1 (C=O), 165.5 (C=O). HRMS (ESI): *m*/*z* calcd. for C_19_H_21_Cl_3_NO_6_^+^ [M + H]^+^: 464.0429, found: 464.0437.

*2,2,2-Trichloroethyl (4aR*,5R*)-7,7-dimethyl-5-phenethyl-4a,5,6,7-tetrahydro-2H-[1,2]oxazino[2,3-b][1,2]oxazine-4-carboxylate* **3m**. Nitroso acetal **3m** was obtained from nitronate **1h** (31 mg, 0.13 mmol) and vinyl diazoacetate **2a** (63 mg, 0.26 mmol) according to GP-1. Column chromatography (eluent: PE/EtOAc, 30:1, then 20:1) afforded 43 mg (72%) of the target nitroso acetal **3m** as colorless oil, which solidified upon storage in a fridge (*ca*. 4 °C). *R*_f_ = 0.47 (PE/EtOAc, 3:1, UV, anisaldehyde). m.p. = 136–138 °C (EtOAc). ^1^H NMR (300 MHz, COSY, CDCl_3_): δ 1.32 (s, 3H, Me(7)), 1.53 (s, 3H, Me(7)), 1.53–1.59 (m, 2H, PhCH_2_CH_2a_ and CH_2ax_(6)), 1.63–1.75 (m, 1H, PhCH_2_CH_2b_), 2.03 (dd, *J* = 13.3, 2.7 Hz, 1H, CH_2eq_(6)), 2.36–2.52 (m, 2H, CH(5) and PhCH_2a_), 2.79 (ddd, *J* = 13.8, 10.6, 5.3 Hz, 1H, PhCH_2b_), 3.87 (app d, *J* = 10.2 Hz, 1H, CH(4a)), 4.49 (dd, *J* = 18.4, 3.1 Hz, 1H, CH_2a_(2)), 4.71–4.81 (m, 2H, OCH_2a_CCl_3_ and CH_2b_(2)), 4.87 (d, *J* = 12.0 Hz, 1H, OCH_2b_CCl_3_), 7.10–7.31 (m, 6H, CH(3) and CH_Ph_). ^13^C NMR (75 MHz, DEPT, HSQC, CDCl_3_): δ 28.9 (Me(7)), 30.4 (CH(5)), 30.8 (Me(7)), 32.6 (PhCH_2_), 33.6 (PhCH_2_CH_2_), 40.5 (CH_2_(6)), 64.5 (CH(4a)), 68.4 (br, CH_2_(2)), 74.1 (OCH_2_CCl_3_), 78.8 (C(7)), 94.8 (CCl_3_), 125.9 (CH_Ph_), 128.3 (CH_Ph_), 128.4 (CH_Ph_), 129.8 (C(4)), 140.5 (=CH(3)), 141.8 (C_Ar_), 163.6 (C=O). HRMS (ESI): *m*/*z* calcd. for C_20_H_25_Cl_3_NO_4_^+^ [M + H]^+^: 448.0844, found: 448.0847.

*2,2,2-Trichloroethyl (4aR*,5S*,7R*)-5,7-diphenyl-4a,5,6,7-tetrahydro-2H-[1,2]oxazino[2,3-b][1,2]oxazine-4-carboxylate* **3n**. Nitroso acetal **3n** was obtained from nitronate **1i** (30 mg, 0.12 mmol) and vinyl diazoacetate **2a** (58 mg, 0.24 mmol) according to GP-1. Column chromatography (eluent: PE/EtOAc, 10:1) afforded 49 mg (88%) of the target nitroso acetal **3n** as colorless oil. Analytically pure material was obtained after crystallization from PE/CH_2_Cl_2_, 3:1 as white powder. *R*_f_ = 0.51 (PE/EtOAc, 3:1, UV, anisaldehyde). m.p. = 128–129 °C (PE/CH_2_Cl_2_, 3:1). ^1^H NMR (300 MHz, COSY, CDCl_3_): δ 2.22 (app dt, *J* = 13.5, 3.2 Hz, 1H, CH_2eq_(6)), 2.48 (app q, *J* = 12.6 Hz, 1H, CH_2ax_(6)), 3.65 (ddd, *J* = 12.5, 10.7, 3.4 Hz, 1H, CH(5)), 3.97 (d, *J* = 11.9 Hz, 1H, OCH_2a_CCl_3_), 4.33 (dd, *J* = 10.7, 1.9 Hz, 1H, CH(4a)), 4.42 (d, *J* = 11.9 Hz, 1H, OCH_2b_CCl_3_), 4.70 (dd, *J* = 18.1, 3.4 Hz, 1H, CH_2a_(2)), 4.96 (dt, *J* = 18.1, 1.8 Hz, 1H, CH_2b_(2)), 5.62 (dd, *J* = 11.8, 2.5 Hz, 1H, CH(7)), 7.02 (dd, *J* = 3.4, 1.8 Hz, 1H, CH(3)), 7.23–7.51 (m, 10H, 2 × Ph). Characteristic NOESY interactions: CH(7)–CH_2eq_(6), CH(7)–CH(5), CH(4a)–CH_2ax_(6), CH(5)–CH_2eq_(6). ^13^C NMR (75 MHz, DEPT, HSQC, CDCl_3_): δ 36.7 (CH_2_(6)), 41.6 (CH(5)), 64.6 (CH(4a)), 68.8 (CH_2_(2)), 73.7 (OCH_2_CCl_3_), 74.1 (CH(7)), 94.7 (CCl_3_), 126.8 (CH_Ph_), 127.4 (CH_Ph_), 128.4 (CH_Ph_), 128.48 (CH_Ph_), 128.51 (CH_Ph_), 128.6 (CH_Ph_), 130.5 (C(4)), 138.7 (=CH(3)), 139.2 (C_Ph_), 139.8 (C_Ph_), 162.7 (C=O). HRMS (ESI): *m*/*z* calcd. for C_22_H_21_Cl_3_NO_4_^+^ [M + H]^+^: 468.0531, found: 468.0527.

*2,2,2-Trichloroethyl (4aR*,5S*,7S*)-7-ethoxy-5-(4-methoxyphenyl)-4a,5,6,7-tetrahydro-2H-[1,2]oxazino[2,3-b][1,2]oxazine-4-carboxylate* **3o**. Nitroso acetal **3o** was obtained from nitronate **1j** (26 mg, 0.10 mmol) and vinyl diazoacetate **2a** (51 mg, 0.21 mmol) according to GP-1. Column chromatography (eluent: PE/EtOAc, 20:1, then 10:1) afforded 34 mg (70%) of the target nitroso acetal **3o** as colorless oil. *R*_f_ = 0.45 (PE/EtOAc, 3:1, UV, anisaldehyde). ^1^H NMR (300 MHz, COSY, CDCl_3_): δ 1.27 (t, *J* = 7.1 Hz, 3H, OCH_2_CH_3_), 2.11 (ddd, *J* = 13.4, 3.3, 1.2 Hz, 1H, CH_2eq_(6)), 2.45 (td, *J* = 13.4, 4.1 Hz, 1H, CH_2ax_(6)), 3.55–3.75 (m, 2H, CH(5) and OCH_2a_CH_3_), 3.79 (s, 3H, OMe), 4.05–4.19 (m, 3H, OCH_2a_CCl_3_, CH(4a), and OCH_2b_CH_3_), 4.37 (d, *J* = 11.9 Hz, 1H, OCH_2b_CCl_3_), 4.55 (dd, *J* = 18.1, 3.5 Hz, 1H, CH_2a_(2)), 4.79 (app dt, *J* = 18.1, 1.8 Hz, 1H, CH_2b_(2)), 5.13 (app d, *J* = 3.2 Hz, 1H, O–CH(7)–O), 6.82 (d, *J* = 8.7 Hz, 2H, CH_Ar_), 6.96 (dd, *J* = 3.0, 1.6 Hz, 1H, CH(3)), 7.18 (d, *J* = 8.7 Hz, 2H, CH_Ar_). Characteristic NOESY interactions: CH(7)/CH_2ax_(6); [CH(5) and OCH_2a_CH_3_]/CH_2eq_(6); CH_2ax_(6)/[OCH_2a_CCl_3_, CH(4a), and OCH_2a_CH_3_]; CH_2ax_(6)/CH_Ar_. ^13^C NMR (75 MHz, DEPT, HSQC, HMBC, CDCl_3_): δ 14.9 (OCH_2_CH_3_), 33.7 (CH(5)), 34.0 (CH_2_(6)), 55.3 (OMe), 63.9 (OCH_2_CH_3_), 65.3 (CH(4a)), 68.7 (CH_2_(2)), 73.8 (OCH_2_CCl_3_), 94.7 (CCl_3_), 100.2 (O–CH(7)–O), 113.7 (CH_Ar_), 129.5 (CH_Ar_), 130.3 (C(4)), 131.7 (C_Ar_), 138.8 (=CH(3)), 158.7 (C_Ar_–OMe), 162.7 (C=O). HRMS (ESI): *m*/*z* calcd. for C_19_H_23_Cl_3_NO_6_^+^ [M + H]^+^: 466.0585, found: 466.0599.

*2,2,2-Trichloroethyl (4aR*,5S*,5aR*,9aR*)-5-(4-chlorophenyl)-4a,5,5a,6,7,8,9,9a-octahydro-2H-benzo[e][1,2]oxazino[2,3-b][1,2]oxazine-4-carboxylate* **3p**. Nitroso acetal **3p** was obtained from nitronate **1k** (30 mg, 0.11 mmol) and vinyl diazoacetate **2a** (56 mg, 0.23 mmol) according to GP-1 (reaction time = 2 days). Column chromatography (eluent: PE/EtOAc, 40:1) afforded 21 mg of crude nitroso acetal as oil, which was crystallized from PE/CH_2_Cl_2_, 3:1 to give 8 mg of target nitroso acetal **3p** as white powder. Column chromatography (eluent: PE/CH_2_Cl_2_, 1:1.5, then 1:2) of filtrate afforded 10 mg of target nitroso acetal **3p** as colorless oil. Total yield: 18 mg (33%). *R*_f_ = 0.47 (PE/EtOAc, 3:1, UV, anisaldehyde). m.p. = 153–155 °C (PE/CH_2_Cl_2_, 3:1). ^1^H NMR (300 MHz, COSY, CDCl_3_, 320 K): δ 1.15–1.38 (m, 4H, CH_2abcd_), 1.57–1.65 (m, 1H, CH_2e_), 1.70–1.79 (m, 1H, CH_2f_), 1.91–2.01 (m, 1H, CH_2a_(9)), 2.31–2.55 (m, 2H, CH(5a) and CH_2b_(9)), 3.60 (app t, *J* = 9.4 Hz, 1H, CH(5)–Ar), 4.22–4.45 (m, 4H, CH(4a)–N, CH(5a)–O, and OCH_2_CCl_3_), 4.59–4.74 (m, 2H, CH_2_(2)–O), 6.99 (br s, 1H, CH(3)=), 7.28 (br s, 4H, CH_Ar_). ^13^C NMR (75 MHz, HSQC, CDCl_3_, 320 K): δ 21.5 (br, CH_2_), 24.7 (br, CH_2_), 27.2 (CH_2_), 31.3 (CH_2_(9)), 38.5 (CH(5a)), 40.0 (br, CH(5)), 64.5 (br, CH(4a)), 67.1 (br, CH_2_(2)), 73.9 (OCH_2_CCl_3_), 78.8 (br, CH(9a)–O), 94.7 (CCl_3_), 128.5 (CH_Ar_), 129.8 (br, C(4)), 130.5 (br, CH_Ar_), 132.8 (C_Ar_), 138.4 (=CH(3)), 162.9 (C=O). C_Ar_ could not be unambiguously identified due to broadening/low intensity/possible overlapping. HRMS (ESI): *m*/*z* calcd. for C_20_H_22_Cl_4_NO_4_^+^ [M + H]^+^: 480.0297, found: 480.0286. The crystallographic information for compound **3p** was deposited in the Cambridge Crystallographic Data Centre (CCDC 2242309).

#### 3.3.2. General Procedure for the Synthesis of Pyrrolooxazines **4** (GP-2)

To the 0.3 M solution of nitronate **1** (1 equiv.) in THF, Rh_2_(Oct)_4_ (0.02 equiv.) was added under an argon atmosphere at r.t. and the mixture was stirred for 5 min. Then THF solution of vinyl diazoacetate **2** (2 equiv. rel. to nitronate **1**, C = 0.5M) was added dropwise (appr. 5–10 min) using a syringe with stirring. The resulting solution was stirred overnight and evaporated. The residue was dissolved in freshly distilled CH_2_Cl_2_ (5 mL/1 mmol of starting nitronate **1**). Then 2,2,2-trichloroethanol (10 equiv.) and DBU (2 equiv.) were consequently added under an argon atmosphere. The resulting solution was maintained for 16–24 h (TLC monitoring) and transferred into EtOAc (20 mL)/H_2_O (10 mL). The organic layer was washed with NaHSO_4_ (0.5 M in H_2_O, 10 mL). Then aqueous layer was washed with EtOAc (10 mL) and the combined layer was washed with brine (15 mL), dried (Na_2_SO_4_), and evaporated. The residue was preadsorbed onto Celite^®^ and subjected to column chromatography on silica gel (eluent: PE/EtOAc or PE/CH_2_Cl_2_) to give target pyrrolooxazines **4**. NB: Even small residual amounts of 2,2,2-trichloroethanol can affect the *R*_f_ values of target pyrroles **4** and chromatography rate.

*2,2,2-Trichloroethyl 4-(4-methoxyphenyl)-2,2-dimethyl-3,4-dihydro-2H-pyrrolo[1,2-b][1,2]oxazine-5-carboxylate* **4a**.

1. To a stirring solution of nitroso acetal **3a** (46 mg, 0.10 mmol) in CH_2_Cl_2_ (0.50 mL), 2,2,2-trichloroethanol (96 μL, 149 mg, 1.00 mmol) and DBU (30 μ, 30 mg, 0.20 mmol) were consequently added under an argon atmosphere. The resulting solution was stirred for 24 h and transferred into EtOAc (20 mL)/H_2_O (10 mL). The organic layer was washed with NaHSO_4_ (0.5 M in H_2_O, 10 mL). Then aqueous layer was washed with EtOAc (10 mL) and the combined layer was washed with brine (15 mL), dried (Na_2_SO_4_), and evaporated. The residue was preadsorbed onto Celite^®^ and subjected to column chromatography on silica gel (eluent: PE/EtOAc, 30:1) to give 32 mg (74%) of the target pyrrolooxazine **4a** as colorless oil, which solidified upon storage in a fridge (*ca*. 4 °C).

2. (One-pot procedure) Pyrrolooxazine **4a** was obtained from nitronate **1a** (26 mg, 0.11 mmol) and vinyl diazoacetate **2a** (54 mg, 0.22 mmol) according to GP-2. Column chromatography (eluent: PE/EtOAc, 50:1, then 40:1) afforded 29 mg (62%) of the target pyrrolooxazine **4a** as colorless oil. *R*_f_ = 0.30 (PE/EtOAc, 5:1, UV, anisaldehyde). m.p. = 93–95 °C (PE/CH_2_Cl_2_, 3:1). ^1^H NMR (300 MHz, CDCl_3_): δ 1.29 (s, 3H, Me), 1.41 (s, 3H, Me), 2.07 (dd, *J* = 14.1, 9.2 Hz, 1H, CH_2a_(3)), 2.32 (dd, *J* = 14.1, 8.6 Hz, 1H, CH_2b_(3)), 3.79 (s, 3H, OMe), 4.50 (d, *J* = 12.0 Hz, 1H, CO_2_CH_2a_CCl_3_), 4.61 (app t, *J* = 8.8 Hz, 1H, CH(4)), 4.79 (d, *J* = 12.0 Hz, 1H, CO_2_CH_2a_CCl_3_), 6.65 (d, *J* = 3.2 Hz, 1H, CH(6 or 7)), 6.72 (d, *J* = 3.2 Hz, 1H, CH(7 or 6)), 6.82 (d, *J* = 8.7 Hz, 2H, CH_Ar_), 7.10 (d, *J* = 8.7 Hz, 2H, CH_Ar_). ^13^C NMR (75 MHz, DEPT, CDCl_3_): δ 22.5 (Me), 27.1 (Me), 35.6 (CH(4)), 43.3 (CH_2_(3)), 55.2 (OMe), 73.1 (OCH_2_CCl_3_), 82.2 (C(2)), 95.8 (CCl_3_), 105.7 (C(5)), 106.5 (CH(6 or 7)), 113.8 (CH_Ar_), 114.3 (CH(7 or 6)), 128.1 (CH_Ar_), 129.9 (C(4a)), 136.5 (C_Ar_), 158.0 (C_Ar_–OMe), 161.9 (C=O). HRMS (ESI): *m*/*z* calcd. for C_19_H_21_Cl_3_NO_4_^+^ [M + H]^+^: 432.0531, found: 432.0520.

*2,2,2-Trichloroethyl 4-(4-chloroxyphenyl)-2,2-dimethyl-3,4-dihydro-2H-pyrrolo[1,2-b][1,2]oxazine-5-carboxylate* **4g**. Pyrrolooxazine **4g** was obtained from nitronate **1b** (27 mg, 0.11 mmol) and vinyl diazoacetate **2a** (54 mg, 0.22 mmol) according to GP-2. Column chromatography (eluent: PE/EtOAc, 40:1) afforded 33 mg (69%) of the target pyrrolooxazine **4g** as white powder. *R*_f_ = 0.42 (PE/EtOAc, 5:1, UV, anisaldehyde). m.p. = 111–113 °C (PE/CH_2_Cl_2_, 3:1). ^1^H NMR (300 MHz, CDCl_3_): δ 1.29 (s, 3H, Me), 1.42 (s, 3H, Me), 2.07 (dd, *J* = 14.1, 9.4 Hz, 1H, CH_2a_(3)), 2.33 (dd, *J* = 14.1, 8.7 Hz, 1H, CH_2b_(3)), 4.53 (d, *J* = 12.0 Hz, 1H, CO_2_CH_2a_CCl_3_), 4.63 (app t, *J* = 9.0 Hz, 1H, CH(4)), 4.77 (d, *J* = 12.0 Hz, 1H, CO_2_CH_2b_CCl_3_), 6.66 (d, *J* = 3.2 Hz, 1H, CH(6 or 7)), 6.74 (d, *J* = 3.2 Hz, 1H, CH(7 or 6)), 7.12 (d, *J* = 8.5 Hz, 2H, CH_Ar_), 7.26 (d, *J* = 8.5 Hz, 2H, CH_Ar_). ^13^C NMR (75 MHz, DEPT, CDCl_3_): δ 22.4 (Me), 27.1 (Me), 35.9 (CH(4)), 43.0 (CH_2_(3)), 73.1 (OCH_2_CCl_3_), 82.1 (C(2)), 95.7 (CCl_3_), 105.9 (C(5)), 106.6 (CH(6 or 7)), 114.6 (CH(7 or 6)), 128.5 (CH_Ar_), 128.6 (CH_Ar_), 128.8 and 132.0 (C(4a) and C_Ar_), 142.8 (C_Ar_), 161.8 (C=O). HRMS (ESI): *m*/*z* calcd. for C_18_H_17_Cl_4_NNaO_3_^+^ [M + Na]^+^: 457.9855, found: 457.9862.

*2,2,2-Trichloroethyl 4-(benzoyloxy)-2,2-dimethyl-3,4-dihydro-2H-pyrrolo[1,2-b][1,2]oxazine-5-carboxylate* **4l**. Pyrrolooxazine **4l** was obtained from nitronate **1g** (27 mg, 0.11 mmol) and vinyl diazoacetate **2a** (54 mg, 0.22 mmol) according to GP-2. Column chromatography (eluent: PE/CH_2_Cl_2_, 1:1, then 1:1.5) afforded 12 mg (25%) of the target pyrrolooxazine **4l** as orange oil. *R*_f_ = 0.19 (PE/CH_2_Cl_2_, 1:2, UV, anisaldehyde). ^1^H NMR (300 MHz, CDCl_3_): δ 1.45 (s, 3H, Me), 1.51 (s, 3H, Me), 2.32 (dd, *J* = 15.3, 3.0 Hz, 1H, CH_2a_(3)), 2.50 (dd, *J* = 15.3, 5.9 Hz, 1H, CH_2b_(3)), 4.61 (d, *J* = 12.0 Hz, 1H, CO_2_CH_2a_CCl_3_), 4.94 (d, *J* = 12.0 Hz, 1H, CO_2_CH_2b_CCl_3_), 6.64 (dd, *J* = 5.9, 3.0 Hz, 1H, CH(4)), 6.72 (d, *J* = 3.2 Hz, 1H, CH(6 or 7)), 6.79 (d, *J* = 3.2 Hz, 1H, CH(7 or 6)), 7.42 (app t, *J* = 7.6 Hz, 2H, CH_Bz_), 7.56 (d, *J* = 7.4, 1.3 Hz, 1H, CH_Bz_), 7.98–8.02 (m, 2H, CH_Bz_). ^13^C NMR (75 MHz, DEPT, CDCl_3_): δ 23.9 (Me), 26.5 (Me), 38.2 (CH_2_(3)), 61.3 (CH(4)), 73.4 (OCH_2_CCl_3_), 81.1 (C(2)), 95.5 (CCl_3_), 107.2 (CH(6 or 7)), 107.9 (C(5)), 115.2 (CH(7 or 6)), 123.5 (C_Ar_), 128.4 (CH_Ar_), 129.7 (CH_Ar_), 130.0 (C(4a)), 133.1 (CH_Ar_), 161.7 (C=O), 165.5 (C=O). HRMS (ESI): *m*/*z* calcd. for C_19_H_18_Cl_3_NNaO_5_^+^ [M + Na]^+^: 468.0143, found: 468.0131.

*2,2,2-Trichloroethyl 2,2-dimethyl-4-phenethyl-3,4-dihydro-2H-pyrrolo[1,2-b][1,2]oxazine-5-carboxylate* **4m**. Pyrrolooxazine **4m** was obtained from nitronate **1h** (26 mg, 0.11 mmol) and vinyl diazoacetate **2a** (54 mg, 0.22 mmol) according to GP-2. Column chromatography (eluent: PE/EtOAc, 70:1) afforded 27 mg (57%) of the target pyrrolooxazine **4m** as colorless oil. *R*_f_ = 0.48 (PE/EtOAc, 5:1, UV, anisaldehyde). ^1^H NMR (300 MHz, COSY, CDCl_3_): δ 1.15 (s, 3H, Me), 1.49 (s, 3H, Me), 1.91–2.04 (m, 2H, PhCH_2_CH_2a_ and CH_2a_(3)), 2.13 (dd, *J* = 13.9, 8.7 Hz, 1H, CH_2b_(3)), 2.39–2.50 (m, 1H, PhCH_2_CH_2b_), 2.65–2.75 (m, 2H, PhCH_2_), 3.58 (app qd, *J* = 8.7, 4.0 Hz, 1H, CH(4)), 4.87–4.95 (m, 2H, OCH_2_CCl_3_), 6.61–6.64 (m, 2H, CH(6) and CH(7)), 7.17–7.23 (m, 2H, CH_Ph_), 7.27–7.32 (m, 3H, CH_Ph_). ^13^C NMR (75 MHz, DEPT, HSQC, CDCl_3_): δ 23.0 (Me), 27.6 (Me), 29.6 (CH(4)), 32.9 (PhCH_2_), 36.9 (PhCH_2_CH_2_), 39.0 (CH_2_(3)), 73.6 (OCH_2_CCl_3_), 82.7 (C(2)), 95.8 (CCl_3_), 104.9 (C(5)), 106.5 (CH(6 or 7)), 114.4 (CH(7 or 6)), 125.9 and 128.4 (all CH_Ph_), 132.2 (C(4a)), 141.9 (C_Ph_), 162.6 (C=O). HRMS (ESI): *m*/*z* calcd. for C_20_H_23_Cl_3_NO_3_^+^ [M + H]^+^: 430.0738, found: 430.0740.

*2,2,2-Trichloroethyl (2R*,4S*)-2,4-diphenyl-3,4-dihydro-2H-pyrrolo[1,2-b][1,2]oxazine-5-carboxylate* **4n**. Pyrrolooxazine **4n** was obtained from nitronate **1i** (28 mg, 0.11 mmol) and vinyl diazoacetate **2a** (54 mg, 0.22 mmol) according to GP-2. Column chromatography (eluent: PE/EtOAc, 70:1, then 50:1) afforded 25 mg (50%) of the target pyrrolooxazine **4n** as colorless oil. *R*_f_ = 0.38 (PE/EtOAc, 5:1, UV, anisaldehyde). ^1^H NMR (300 MHz, CDCl_3_): δ 2.41 (ddd, *J* = 14.4, 11.8, 9.8 Hz, 1H, CH_2a_(3)), 2.76 (ddd, *J* = 14.4, 8.9, 1.9 Hz, 1H, CH_2b_(3)), 4.36 (d, *J* = 12.0 Hz, 1H, CO_2_CH_2a_CCl_3_), 4.85 (d, *J* = 12.0 Hz, 1H, CO_2_CH_2b_CCl_3_), 4.90 (app t, *J* = 9.2 Hz, 1H, CH(4)), 5.31 (dd, *J* = 11.8, 1.9 Hz, 1H, CH(2)), 6.69 (d, *J* = 3.3 Hz, 1H, CH(6 or 7)), 6.85 (d, *J* = 3.3 Hz, 1H, CH(7 or 6)), 7.20–7.34 (m, 5H, Ph), 7.40–7.46 (m, 5H, Ph). ^13^C NMR (75 MHz, DEPT, CDCl_3_): δ 38.9 (CH(4)), 39.7 (CH_2_(3)), 72.8 (OCH_2_CCl_3_), 84.2 (CH(2)), 95.8 (CCl_3_), 106.2 (C(5)), 106.7 (CH(6 or 7)), 114.0 (CH(7 or 6)), 126.5 (CH_Ph_), 126.86 (CH_Ph_), 126.93 (CH_Ph_), 128.6 (CH_Ph_), 128.9 (CH_Ph_), 129.4 (CH_Ph_), 129.5 (C(4a)), 136.7 (C_Ph_), 144.5 (C_Ph_), 161.8 (C=O). HRMS (ESI): *m*/*z* calcd. for C_22_H_18_Cl_3_NNaO_3_^+^ [M + Na]^+^: 472.0244, found: 472.0234.

*2,2,2-Trichloroethyl (2S*,4S*)-2-ethoxy-4-(4-methoxyphenyl)-3,4-dihydro-2H-pyrrolo[1,2-b][1,2]oxazine-5-carboxylate* **4o**. Pyrrolooxazine **4o** was obtained from nitronate **1j** (18 mg, 0.07 mmol) and vinyl diazoacetate **2a** (34 mg, 0.14 mmol) according to GP-2. Column chromatography (eluent: PE/EtOAc, 50:1, then 30:1) afforded 15 mg (47%) of the target pyrrolooxazine **4o** as orange oil. *R*_f_ = 0.31 (PE/EtOAc, 5:1, UV, anisaldehyde). ^1^H NMR (300 MHz, CDCl_3_): δ 1.22 (t, *J* = 7.1 Hz, 3H, OCH_2_CH_3_), 2.30–2.47 (m, 2H, CH_2_(3)), 3.67 (dq, *J* = 9.7, 7.1 Hz, 1H, OCH_2a_CH_3_), 3.79 (s, 3H, OMe), 3.91 (dq, *J* = 9.7, 7.1 Hz, 1H, OCH_2b_CH_3_), 4.54 (d, *J* = 12.0 Hz, 1H, CO_2_CH_2a_CCl_3_), 4.60 (app t, overlapped, *J* = 7.2 Hz, 1H, CH(4)), 4.81 (d, overlapped, *J* = 12.0 Hz, 1H, CO_2_CH_2b_CCl_3_), 5.35 (app t, *J* = 4.0 Hz, 1H, O–CH(7)–O), 6.66 (d, *J* = 3.3 Hz, 1H, CH(6 or 7)), 6.73 (d, *J* = 3.3 Hz, 1H, CH(7 or 6)), 6.83 (d, *J* = 8.7 Hz, 2H, CH_Ar_), 7.09 (d, *J* = 8.6 Hz, 2H, CH_Ar_). ^13^C NMR (75 MHz, DEPT, CDCl_3_): δ 15.0 (OCH_2_CH_3_), 34.2 (CH(4)), 36.2 (CH_2_(3)), 55.2 (OMe), 65.2 (OCH_2_CH_3_), 73.2 (OCH_2_CCl_3_), 95.7 (CCl_3_), 102.1 (O–CH(7)–O), 105.6 (C(5)), 106.6 (CH(6 or 7)), 113.9 (CH_Ar_), 114.1 (CH(7 or 6)), 128.2 (CH_Ar_), 130.8 (C(4a)), 135.6 (C_Ar_), 158.1 (C_Ar_–OMe), 162.0 (C=O). HRMS (ESI): *m*/*z* calcd. for C_19_H_21_Cl_3_NO_5_^+^ [M + H]^+^: 470.0299, found: 470.0295.

#### 3.3.3. Preparation of Pyrrolooxazines **4b**, **5**

*Methyl 4-(4-methoxyphenyl)-2,2-dimethyl-3,4-dihydro-2H-pyrrolo[1,2-b][1,2]oxazine-5-carboxylate* **4b**. To a stirring solution of nitroso acetal **3a** (31 mg, 0.07 mmol) in a mixture of MeOH (0.34 mL) and MeCN (0.17 mL), DBU (20 μL, 20 mg, 0.13 mmol) was added dropwise under an argon atmosphere at r.t. The resulting solution was left overnight and transferred into EtOAc (20 mL)/H_2_O (10 mL). The organic layer was washed with NaHSO_4_ (0.5 M in H_2_O, 10 mL). Then aqueous layer was washed with EtOAc (10 mL) and the combined layer was washed with brine (15 mL), dried (Na_2_SO_4_), and evaporated. The residue was preadsorbed onto Celite^®^ and subjected to column chromatography on silica gel (eluent: PE/EtOAc 20:1, then 12:1) to give 13 mg (60%) of the target pyrrolooxazine **4b** as colorless oil, which solidified upon storage in a fridge (*ca*. 4 °C). *R*_f_ = 0.29 (PE/EtOAc, 3:1, UV, anisaldehyde). mp = 93–95 °C (PE/CH_2_Cl_2_, 3:1). ^1^H NMR (300 MHz, COSY, CDCl_3_): δ 1.25 (s, 3H, Me), 1.41 (s, 3H, Me), 2.05 (dd, *J* = 14.1, 9.5 Hz, 1H, CH_2a_(3)), 2.28 (dd, *J* = 14.1, 8.7 Hz, 1H, CH_2b_(3)), 3.51 (s, 3H, CO_2_Me), 3.79 (s, 3H, OMe), 4.54 (app t, *J* = 9.1 Hz, 1H, CH(4)), 6.53 (d, *J* = 3.2 Hz, 1H, CH(6 or 7)), 6.68 (d, *J* = 3.2 Hz, 1H, CH(7 or 6)), 6.83 (d, *J* = 8.7 Hz, 2H, CH_Ar_), 7.10 (d, *J* = 8.7 Hz, 2H, CH_Ar_). ^13^C NMR (75 MHz, DEPT, HSQC, HMBC, CDCl_3_): δ 22.2 (Me), 27.2 (Me), 35.8 (CH(4)), 43.6 (CH_2_(3)), 50.5 (CO_2_Me), 55.2 (C_Ar_–OMe), 82.0 (C(2)), 105.9 (CH(6 or 7)), 107.4 (C(5)), 113.8 (CH_Ar_), 113.9 (CH(7 or 6)), 128.0 (CH_Ar_), 128.7 (C(4a)), 137.2 (C_Ar_), 157.9 (C_Ar_–OMe), 164.5 (C=O). HRMS (ESI): *m*/*z* calcd. for C_18_H_22_NO_4_^+^ [M + H]^+^: 316.1543, found: 316.1550. The crystallographic information for compound **4b** was deposited in the Cambridge Crystallographic Data Centre (CCDC 2242308).

*4-(4-Methoxyphenyl)-2,2-dimethyl-3,4-dihydro-2H-pyrrolo[1,2-b][1,2]oxazine-5-carboxylic acid* **5**. To a stirring solution of nitroso acetal **3a** (35 mg, 0.08 mmol) in THF (0.39 mL), *t*-BuONa (15 mg, 0.16 mmol) was added under an argon atmosphere at r.t. The resulting solution was stirred for 0.5 h and transferred into EtOAc (20 mL)/H_2_O (10 mL). The organic layer was washed with NaHSO_4_ (0.5 M in H_2_O, 10 mL). Then aqueous layer was washed with EtOAc (10 mL) and the combined layer was washed with brine (15 mL), dried (Na_2_SO_4_), and evaporated. The residue was preadsorbed onto Celite^®^ and subjected to column chromatography on silica gel (eluent: PE/EtOAc, 20:1, then 5:1, then 3:1) to give 2 mg (6%) of pyrrolooxazine **4a** and 5.5 mg (23%) of the target pyrrolooxazine acid **5** as an oil, which solidified upon storage in a fridge (*ca*. 4 °C) as white powder. *R*_f_ = 0.28 (PE/EtOAc, 1:1, UV, anisaldehyde). m.p. = 185–187 °C (PE/CH_2_Cl_2_, 3:1). ^1^H NMR (300 MHz, CDCl_3_): δ 1.22 (s, 3H, Me), 1.41 (s, 3H, Me), 2.09 (dd, *J* = 14.2, 8.7 Hz, 1H, CH_2a_(3)), 2.30 (dd, *J* = 14.2, 8.9 Hz, 1H, CH_2b_(3)), 3.79 (s, 3H, OMe), 4.58 (app t, *J* = 8.8 Hz, 1H, CH(4)), 6.54 (d, *J* = 3.2 Hz, 1H, CH(6 or 7)), 6.68 (d, *J* = 3.2 Hz, 1H, CH(7 or 6)), 6.81 (d, *J* = 8.7 Hz, 2H, CH_Ar_), 7.09 (d, *J* = 8.7 Hz, 2H, CH_Ar_). ^13^C NMR (75 MHz, DEPT, CDCl_3_): δ 22.5 (Me), 27.2 (Me), 35.5 (CH(4)), 43.4 (CH_2_(3)), 55.2 (OMe), 82.5 (C(2)), 106.4 (C(5)), 106.5 (CH(6 or 7)), 113.8 (CH_Ar_), 114.3 (CH(7 or 6)), 128.3 (CH_Ar_), 130.3 (C(4a)), 136.8 (C_Ar_), 157.9 (C_Ar_–OMe), 168.3 (C=O). HRMS (ESI): *m*/*z* calcd. for C_17_H_20_NO_4_^+^ [M + H]^+^: 302.1384, found: 302.1390.

## 4. Conclusions

In conclusion, an approach was developed for the synthesis of bicyclic nitroso acetals (4a,5,6,7-tetrahydro-2*H*-[1,2]oxazino[2,3-*b*][1,2]oxazines) via an Rh(II)-catalyzed annulation of six-membered cyclic nitronates with vinyl diazoacetates. Target products were obtained in good yields with excellent diastereoselectivity. These nitroso acetals undergo smooth ring contraction under the action of the DBU/alcohol system. Combined with a [3+3]-annulation in a one-pot protocol, it represents a quick approach to the assembly of pyrrolooxazines and complements already known approaches to this relatively rare heterocyclic core.

## Data Availability

The data presented in this study are available in this article and in the Appendix A.

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
