# Peer review of "[3+3]-Annulation of Cyclic Nitronates with Vinyl Diazoacetates: Diastereoselective Synthesis of Partially Saturated [1,2]Oxazino[2,3-b][1,2]oxazines and Their Base-Promoted Ring Contraction to Pyrrolo[1,2-b][1,2]oxazine Derivatives"

_molecules, 2023, doi:10.3390/molecules28073025_

Round 1

Reviewer 1 Report

This paper describes the reaction between model nitronate and vinyl diazoacetate. In cyclic nitronates with vinyl diazoacetates, [3+3] type annulation pathway is rare. Scope of substrates is well examined. Diastereoselective reaction is interesting. Products were well characterized including X-ray analysis and 2D NMR.

From these reasons, I conclude this paper is suitable for publication in Molecules. Before publication please consider the following a slight point.

1. About proposed mechanism, it seems plausible. As an another pathway, can not be an inversed pathways like initial attack of Rh carbenoid to C=N bond of nitrones followed by oxygen attack to terminal C=C bond? 

2. In experimental section: Please add the information of substrates 1 such as availability, preparation method, references.

Author Response

  • About proposed mechanism: Carbenes and metallocarbenes are known as electrophilic species (see, Refs.42,43,62 and many others, e.g. Chem. Rec. 2017, 17, 312-325). Their formation from vinyl diazo compounds is considered as a reactivity umpolung allowing transformation of formal nucleophile (vinyl diazo) into and electrophile (vinyl carbenoid). In turn, nitronates 1 are rather good nucleophiles due to a N-oxide moiety. To increase the electrophilicity of nitronates, their transformation into iminium species (intermediate B in our case, Scheme 3) is required (see e.g., Chem. Rec. 2018, 18, 1489-1500). Therefore, we suppose the first reaction step to be the interaction of nitronate (nucleophile) and carbenoid (electrophile). That is in accordance with mentioned reactivity patterns. To underline the electrophilic character or intermediates A, the sentence was revised as “At the first step electrophilic vinyl-carbenoid species A are generated from vinyl diazo compounds and a rhodium catalyst.”
  • About preparation of substrates in Experimental section: The sentence: “Starting nitronates 1 [83] and vinyl diazoacetates 2 [84] were prepared according to the literature (see Supplementary materials for detailed procedures).” Appropriate Refs. 83 and 84 were added to Reference list.

Reviewer 2 Report

The manuscript of Tabolin and coworkers describes a Rh-catalyzed [3+3] annulation reaction of cyclic nitronates with vinyl diazoacetates. The reaction proceeds through a Rh-carbene intermediate generated from the Rh(II) catalyst and the diazoacetate, furnishing the observed cyclo-adducts in moderate to good chemical yields. Furthermore, the thus-obtained products could also be effortlessly transformed into substituted pyrroles. This synthetic usefulness of this Rh(II)-catalyzed reaction to expediently access products of cyclic nitroso acetals is recommended. Still, this reviewer would like to suggest that the authors prepare scheme 4 in a more detailed fashion, as it is used to explain the mechanistic pathway of forming compound 4.

Author Response

About mechanism in Scheme 4: Scheme 4 was modified. Particularly, C(2) was marked in substrate 3, hydrogen bonding that is proposed to facilitate formation of target product 4 from intermediate E was shown. Also, more detailed discussion was added to the text.

Reviewer 3 Report

1. In the abstract section, I suggest the author extend your design and final results in detail.

2. I suggest the authors have to highlight your design results, what’s the further exploration when compared your previous report.

2. Some related refs could be cited, such as Org. Chem. Front., 2020,7, 3515-3520; Chem. Commun., 2022, 58, 6653–6656; Molecules, 2019, 24, 1760 and J. Org. Chem. 2019, 84, 14627−14635

3. Source and purity of all chemicals used should be specified in the experimental section.

4. Could you do the DFT method for supporting the current mechanism?

5. The author could discuss the structure feature and intramolecular interaction.

6. The title does not highlight the highlights of the draft well.

Author Response

  1. - About abstract: More details were added to the Abstract.
  2. - About comparison with previous data: Introduction was expanded. More details were added to the previously reported [3+3]-annulations of nitronates 1 and known annulations of vinyl diazoacetates 2.
  3. - About more references: We looked through these papers (doi numbers: 10.1039/D0QO01092H; 10.1039/D2CC00146B; 10.3390/molecules24091760; 10.1021/acs.joc.9b02211). First of them is about transition metal-free reduction of amides into primary amines with HBpin. The second one – about asymmetric alkylation of acetoacetates with in situ generated p-quinone methides using copper bis-oxazoline ligands. The third paper is about pyrrolepyrrole-thiophene-containing compounds useful for photovoltaics. Target compounds were prepared by palladium cross-coupling. The 4th paper is about transition metal-free reduction of amides into primary amines with silanes. As we can conclude, these papers are not about oxazines, nitronates, or diazo compounds. There are not rhodium(II) catalysis described. We suppose that they are not relevant to the presented manuscript and should not be cited here.
  4. - About sources of chemicals: Sources (Sigma-Aldrich, Acros, AlfaAesar, etc.) and purities of reagents were added to General Experimental.
  5. - About DFT: DFT calculations are time- and resource-consuming, especially when heavy atom (rhodium in our case) is present and total number of atoms is large. Additional issues arise when considering all possible conformations for both nitronate and metal-carbenoid. For example, [3+2]-cycloaddition of nitrones with vinyl diazo acetates was the topic of special publications: J. Org. Chem. 2016, 81, 8082−8086, DOI: 10.1021/acs.joc.6b01447; RSC Adv. 2016, 6, 53839–53851; 10.1039/c6ra07873g). In the present manuscript the mechanistic explanation is based on both experimental and literature data. We believe that proposed mechanism is reasonable and detailed DFT is not required now.
  6. - About structural features: Apart from already discussed assignment of relative configuration of products 3 and explanation of high diastereoselectivity of [3+3]-annulation, we added the following discussion: “Notably, cis-junction of two oxazine rings was observed, that can be favored by anomeric interaction within O–N–O moiety [61].” Regarding possible other interactions: according to X-ray analysis of products 3b, 3p, and 4b there are no specific intramolecular interactions which deserve discussion.
  7. - About the Title: To highlight both [3+3]-annulation and synthesis of pyrrolo-oxazines, the Title was revised as: “[3+3]-Annulation of Cyclic Nitronates with Vinyl Diazoacetates. Diastereoselective Synthesis of Partially Saturated [1,2]Oxazino[2,3-b][1,2]oxazines and Their Base-Promoted Ring Contraction to Pyrrolo[1,2-b][1,2]oxazine Derivatives”